# Oral Health Needs and Barriers among Children in Saudi Arabia

**DOI:** 10.3390/ijerph192013584

**Published:** 2022-10-20

**Authors:** Heba Jafar Sabbagh, Sarah Abdullah Aljehani, Bayan Mohammed Abdulaziz, Nada Zafer Alshehri, Maryam Omar Bajkhaif, Shatha Khalid Alrosini, Reham Mohammed Al-Amoudi, Heba Mohamed Elkhodary

**Affiliations:** 1Pediatric Dentistry Department, Faculty of Dentistry, King Abdulaziz University, Jeddah 22254, Saudi Arabia; 2Faculty of Dentistry, King Abdulaziz University, Jeddah 22254, Saudi Arabia; 3Pediatric Dentistry Department, Taif Dental Specialty Center, Ministry of Health, Taif 26514, Saudi Arabia; 4Department of Pedodontics and Oral Health, Faculty of Dental Medicine for Girls, Al Azhar University, Cairo 11651, Egypt

**Keywords:** oral health needs, oral health barriers, accessibility, children, Saudi Arabia

## Abstract

(1) Background: Understanding oral health needs and barriers is important to overcome the oral disease burden, especially after the COVID-19 pandemic. The aim of this study was to assess oral health needs and barriers among Saudi children after the COVID-19 pandemic wave started. (2) Methods: Parents of healthy children aged 3–11 years from five regions of Saudi Arabia were surveyed using an electronically administered validated questionnaire from Febuary-2021 to July-2021. Oral lesions/manifestations during the last 12 months reported by parents included tooth discoloration, ulcers, abscess, swelling of the gingiva, halitosis, gingival bleeding, dry mouth, pain while eating, difficulty in speaking or eating, burning sensation, and white spots. Barriers were assessed according to the WHO Oral Health Questionnaire. (3) Results: Children experiencing toothaches were reported by 1098 (72.4%) respondents. When reported, oral manifestations and lesions were associated with a higher inability to visit a dentist (*p* < 0.001). Barriers to dental care were more frequent among parents with lower education (*p* < 0.001; adjusted-odds ratio (AOR), 1.815) and a greater number of children (*p* < 0.001; AOR, 1.197). (4) Conclusion: Parents reported a high frequency of unmet oral health needs. Education could play an important role in improving oral health care in children and as a predictor of public health concerns.

## 1. Introduction

Oral diseases, dominated by dental caries, are a global health concern because they affect more than half of the population [1,2,3]. Other oral diseases, such as oral lesions and manifestations, including signs and symptoms of oral or systemic diseases, can negatively affect children and adolescents and impact their quality of life [4,5]. They may cause pain, discomfort, acute and chronic infections, eating and sleeping disturbances, a higher risk of hospitalization, and loss of school days [6]. Surgeon General Report on Oral Health reported that oral health is a fundamental part of general health and plays a significant role in the overall improvement in well-being in a person’s life [7]. Therefore, it is important to evaluate the extent of oral diseases and the presence of oral manifestations to establish an adequate oral health care setting to understand children’s oral health care needs, provide routine dental visits, and address the barriers to adequate oral health care [8,9,10,11].

Although studies have previously assessed accessibility and oral health care needs, the COVID-19 pandemic has affected accessibility and created barriers that did not exist before [12,13]. Reports showed that >95% of dental facilities were completely closed or open only for emergency care [14,15]. Additionally, the geographical and social variations between different countries and even nationally between different regions show different barriers and availability of care. It could be expected that rural populations find it more difficult access to oral health care than urban populations [16]. Understanding barriers and providing adequate accessibility are among the founding principles of the National Health Service [17].

In Saudi Arabia, the burden of dental caries is high and has increased in the last decades, with an estimated prevalence of 70% among children [18]. Research suggests that multiple risk factors cause delays in seeking dental care, such as parental beliefs, practices, lack of economic resources, and accessibility to dental services [19]. Since dental services provided by the Ministry of Health are free of charge, financial issues should not be a barrier to receiving treatment from public facilities. Therefore, issues such as long waiting times and the location of facilities could be factors affecting accessibility [20,21].

However, to date, there has been a lack of data that can be generalized to the entire community of Saudi Arabia. Hence, the present national study aimed to assess oral health needs and dental barriers among children aged 3–11 years after the start of the COVID-19 pandemic in Saudi Arabia.

## 2. Materials and Methods

### 2.1. Subjects

This study was approved by the Research Ethics Committee of Faculty of Dentistry, King Abdulaziz University (approval No. 232-03-21/30 May 2021). The study was carried out from February 2021 to July 2021 and included parents of healthy children aged 3–11 years from five regions of Saudi Arabia (south, north, west, south, and central region). The suggested sample size for each region was 138 subjects, measured using the OpenEpi (Version 3) online calculator [22], with 95% confidence interval and a suggested prevalence of 29.2%. This suggested prevalence of difficulty in accessing dental care, was suggested according to Elkhodary et al. (2022) [12]. The inclusion criterion was parents of healthy children aged 3–11 years living in Saudi Arabia. Parents of medically compromised children aged <3 years or >11 years were excluded from this study.

### 2.2. Methods

An electronically administered parental questionnaire was structured and validated in English and Arabic by five experts with content validity index (CVI) scores of 0.94 and 0.98, respectively, and face validity was assessed with 10 parents to ensure clarity. It started with a consent form, which was approved by parents. The questionnaire included questions from the domains indicated below.

Part I: General information, including data and sociodemographic factors such as region of residence, parental education level, child age, child sex, and child order between siblings.

Part II: Oral health needs, including the oral health status of the child described by the parents and any oral discomfort/lesions or challenges related to oral health that the child had encountered during the last 12 months according to the WHO Oral Health Questionnaire for children [23]. Oral lesions and manifestations included tooth discoloration (with or without caries), ulcers, abscess, swelling of the gingiva, halitosis, gingival bleeding, dry mouth, pain while eating, difficulty in speaking or eating, burning sensation, and white spots.

Part III: Barriers to oral health care needs, including frequency and reasons for dental visits, any previous dental treatment, barriers to treatment such as fear of COVID-19, distance from the dental clinic, expense of an appointment due to health ban, difficulty of transportation, responsibilities towards family members, not being familiar with appointment booking, and no available appointments.

The questionnaire was distributed electronically using the snowball sampling technique. In order to improve generalizability, we sent the questionnaire through multiple routes. Initially, we assigned a representable data collector from each region to insure distributing the sample among the different regions of Saudi Arabia. Then, we sent the questionnaire through different community members, such as teachers, workers, and primary health care service centers that provided health care for the community

The data were recruited and analyzed using SPSS software, version 23 (IBM Corp., Armonk, NY, USA). Categorical variables were presented as frequencies and percentages and were compared using the chi-square test. Continuous variables were presented as means and standard deviations and were compared using the *t*-test. A binary regression analysis was conducted to identify factors predicting barriers to dental care. Parents reporting barriers to dental care were entered as a dependent factor. Sociodemographic factors including region, parental education, child gender and age, and number of children were entered as independent factors. The significance level was set at *p* < 0.05.

## 3. Results

The questionnaire was sent to 1722 parents in Saudi Arabia. Out of them, 1516 completed the questionnaire, with an 88.3% response rate. The highest proportions of participants were recorded in the central region (477 (31.5%)) and the western region (463 (30.5%)). Children of 784 (51.7%) parents were boys, and 732 (48.3%) were parents of girls; in total, 567 (38.0%) were parents of preschool children aged 3–5 years. Parents of 374 (24.7%) children reported that during the last 12 months, their children had required dental treatment, but they were unable to visit a dentist (Table 1).

Although 384 (25.3%) parents reported excellent oral health in their children, 140 (9.2%) reported poor oral health. Moreover, the highest prevalence of parents reporting inability to visit a dentist were among those reporting good (162 (43.3)) or bad oral health (83 (22.2%)). In contrast, the lowest prevalence was reported by those reporting excellent oral health (28 (7.5%)) among their children, with the difference being statistically significant (*p* < 0.001). Moreover, while 377 (24.9%) parents reported that their children had never had toothache during the last 12 months, 1098 (72.4%) reported the experience of a toothache, ranging from always (76 (5.0%)) to sometimes (487 (32.1%)) and rarely (535 (35.3%)). Moreover, 45 (12.0) or 194 (51.9) had always or sometimes experienced a toothache but were unable to visit the dentist, respectively.

Respondents observing oral manifestations/lesions in their children’s oral cavity had a greater inability to visit the dentist (250 (66.8%)) than those observing no oral manifestations/lesions (124 (33.2%)), with a significant difference (*p* < 0.001). The most commonly observed oral manifestations reported by parents were bad breath (231 (15.2%)), tooth discoloration (194 (12.8%)), and pain while eating (175 (11.5%)) (Table 2).

The binary regression analysis showed that inability to visit dental clinics when needed was significantly greater among children with lower paternal education (*p* < 0.001; adjusted odds ratio (AOR), 1.815; 95% confidence interval (CI), 1.368– 2.409) and those whose parents had a greater number of children (*p* < 0.001; AOR, 1.197; 95% CI, 1.084–1.322). In contrast, significantly fewer younger children (aged 3–5 years) were unable to reach dental clinics (*p* = 0.004; AOR, 0.631; 95% CI, 0.462–0.862). There were no significant associations among child sex, maternal education, region of residence, and parents reporting inability to reach dental facilities (Table 3).

The most common reason reported by parents for the inability of the child to visit the dentist despite the child’s need was the expense of the appointment (156 (9.9%)); inability to make an appointment (154 (9.8%)) and fear of COVID-19 (103 (6.6%)) were the other reasons (Table 4).

## 4. Discussion

Maintaining healthy primary dentition is essential for a child’s overall oral and general development. Parents and family members are considered the primary source of a child’s health and habits, which have an intimate association in determining a child’s oral hygiene status, achieving the best oral health outcomes, and assuring their well-being. In this study, an electronically administered questionnaire was sent to 1722 parents of children aged 3–11 years from five different regions in Saudi Arabia to assess dental barriers and oral health needs among young children. In this study, a quarter of the participants reported that during the last 12 months, their children had required dental treatment but could not visit a dentist. This was consistent with a study in Jeddah, which found that one in four children had never visited a dentist and one in five could not receive dental care when needed in the past 12 months [24]. Similarly, a study conducted by Bahannan et al. [25] found that only 6.8% of participants visited a dentist every 6–12 months, and 45.8% of participants visited their dentist only when needed, while 44.6% never visited a dentist. This was owing to the lack of oral health knowledge among these participants [25]. Another study conducted in the eastern province of Saudi Arabia reported that 38% of school children had never visited a dentist. The results indicated that the present dental needs of Saudi schoolchildren are much higher than those in developed nations [26], which indicates the necessity of understanding the main perceived barriers to accessing needed dental care. Furthermore, this study considered the different regions of Saudi Arabia, and the different regions comprised different population sizes, with the western and central regions covering the highest population frequency (more than 75% of the Saudi population) [27]. This was suggested to affect the accessibility, demand for, and burden of oral health care among the different regions [28,29]. In addition, the burden on parents included more difficult transportation and diverse life style in metropolitan cities compared with small cities [30,31].

Dental needs are a necessary and adequate component for analyzing the demand for dental treatment and the utilization of dental services [32]. Our study showed that parents of children with poor oral health, toothache during the last 12 months, and oral lesions reported significantly more difficulty in getting their children to the necessary dental appointments than the parents of children with excellent oral health or those whose children had never experienced toothaches or oral lesions. The findings were similar to a study conducted in a rural area of Asser province in Saudi Arabia reporting that 54.6% of children’s visits to the dentist were for a toothache [26], reflecting the high prevalence of dental caries in the Saudi population. Similarly, Al Agili and Farsi et al. [24] reported that dental caries (38.4%) and toothache (43.8%) were the most commonly reported reasons for a past dental visit. The results were contradictory with respect to the findings of a similar study in young children in the eastern part of the country, where dental needs or parents’ reports of toothache in their children in the past 6 months were not associated with dental service use after controlling for other variables [24]. Globally, a comparable study from Karachi reported that 31.7% of the participants had visited a dentist; among them, 26% of children visited the dentist only when they had a dental problem, whereas 69.3% of children never visited a dentist [26]. In China, Gao et al. mentioned that parents who evaluated their child’s dental status as “poor” took their child to the dentist more often than those who evaluated their child’s status as “good” [33].

A binary logistic regression analysis was performed to identify the factors influencing the underutilization of dental services and various barriers to dental care access. Our study revealed that young children, paternal education, and families with more children were significant predictors of the inability to visit the dentist when needed. Among the most at-risk and underserved groups are young children who are unable to communicate effectively [8]. This supports our study results, with parents of older children reporting more difficulty in accessing needed dental care than parents of younger children. As children grow older, their odds of complaining about the need of a dental visit increase because they are more capable of communicating and expressing their requirements and needs verbally and have demands for dental care that go beyond caries treatment and prevention [32]. However, the decreased odd ratio for dental barriers for younger children could also be attributed to the better accessibility and appointment availability for younger children in the health care system reported by parents in a previous study conducted on medical health care barriers [28].

In addition to widely available dental care throughout the country, education is a significant risk factor for the use of dental services by young children [24]. Although the study did not prove a significant direct relationship between maternal education and access to dental care, paternal education was found to be a significant predictor for children’s inability to reach dental clinics when needed. Interestingly, we found that fathers with high school education were significantly associated with not being able to access the needed dental care compared with those with higher education. This might be because parents with lower education might be paid daily, and taking their child to the dentist might lead to the loss of their salaries or difficulty in arranging their working hours besides their oral health awareness compared with those with higher education [34].

This is supported by the findings of Hamasha et al. The lack of perceived need and dental insurance, high expenses, transportation, and fear of dental treatment were the most common significant barriers to dental services [35]. These findings were supported by those of Almutlaqah et al. and El Bcheraoui et al., who found that the participants’ intention to use dental services increased with education [8,36]. The results were in contrast with the study by Obeidat et al. [37], in which demographic factors including age and educational level did not show any significant difference with regard to the utilization and regularity of dental services. This might have been due to social differences between the studied populations in Jordan and Saudi Arabia, as well as variances in cultural and economic aspects, in addition to subjectivity in answering the questions.

Parents with more children had greater difficulty in obtaining the necessary dental care for their children. This could be explained by the fact that managing more children by mothers makes it difficult to tend to all their needs, especially for employed mothers [38]. Another report also suggested an increased caries risk among the children of employed mothers [39]. Similar findings were reported by Kotha et al. [40] although the results were not statistically significant.

Moreover, although not significant, the AOR was increased in the northern region with a smaller population size than that of the central region, which includes the capital city and a larger population size [27]. This was also supported by a previous study conducted to assess the accessibility of medical care in children reporting more barriers for parents in rural areas than for parents in urban areas [28].

When parents were asked about the reason for not being able to take their children to the dentist despite their needs, the most cited reason was the cost of the appointment, followed by a lack of available appointments. Although dental care is delivered almost free of charge at Saudi Arabian Ministry of Health institutions across the country, some believe that patients in privately funded care receive more time and better attention from doctors [41]. Others might prefer dental care from private clinics over government clinics because of the availability of various treatment options, the quality of dental care, the ease and promptness of appointment scheduling, the absence of waiting times, and opportunity to continue treatment [8]. This might elucidate the findings of our study, although the study participants’ preference for receiving the required dental care in private dental offices as opposed to governmental clinics was not examined in the current research study. These findings were in line with those of Alagili and Farsi [24], who claimed that 40% of parents whose children needed dental treatment were unable to obtain it because of issues with the system for delivering oral health care, including difficulties in scheduling an appointment. In contrast to our findings, a study conducted in India in a dental college hospital, where services were provided at a fair cost compared with private clinics, did not find the cost of treatment to be a significant barrier [42]. The fear of COVID-19 was another reason why parents were unable to take their children to a dental clinic. It was reported that mothers generally thought that dental offices provided a risk for SARS-CoV-2 transmission [43]; thus, the majority only took their children in an emergency. This was also supported by the fact that dental appointments for kids had decreased during the pandemic compared with visits before it [13]. However, our study results showed that 93.4% the parents were not afraid to visit the dentist with their children, which was inconsistent with the study results by Farsi and Farsi, which were conducted early in the pandemic (June 2020) and reported that 24% of parents refused to take their children to their appointments during the pandemic [43]. This could indicate that parents, over time, developed confidence in the health care system and their infection control commitment. This could have resulted from the fact that Saudi Arabia was one of the best countries that followed the guideline against COVID-19 infection spread during the pandemic [44,45].

The results of the current study were difficult to generalize to the population of Saudi Arabia owing to the snowball sampling technique, which might have limited the ability to gain access to the group of choice, and internet access was required to be involved in the study, leading to selection bias. However, the methodological aspects of this study included the sound internal and external validity of the study design, the distribution of the sample among the different regions, the reliability and validity of the questionnaire, sound study design, and appropriate statistical analyses. In addition, we took care to distribute the questionnaire in the five regions of Saudi Arabia to improve generalizability. Another limitation of this study was the possibility of self-reporting bias. Moreover, data were collected during the COVID-19 pandemic, which may have affected the accessibility of dental care for participants in the study.

## 5. Conclusions

Parents reported a high frequency of unmet oral health needs. Access to dental care was found to be a multidimensional issue, along with education, which was considered a significant predictor. The predominant barriers to the utilization of dental services and inability to visit the dentist despite the child’s needs were the cost of the appointment, the inability to make an appointment, and the fear of COVID-19.

## 6. Clinical Implementation

As illustrated in this study, poor oral hygiene was highly prevalent among children who were unable to visit dental clinics or attend dental appointments; thus, activating mobile clinics to ensure access to dental care in rural areas could play a critical role in improving oral health.

In addition, the focus should be on educating well-qualified primary care dentists who desire to address the challenges of providing care in rural and underserved areas and visit schools for primary oral health awareness.

## 7. Recommendations

There are several gaps in our knowledge regarding barriers to dental care because it depends on cultural and public perceptions, and we would benefit from further research, including realistic evaluation by developing a simpler questionnaire and combining it with a clinical examination. In addition, this study stresses the importance of appointment availability at clinics enrolled in a free dental health care system to improve dental health care accessibility.

## Figures and Tables

**Table 1 ijerph-19-13584-t001:** Participant characteristics distributed according to a parental report of their children’s barriers to oral health care (N = 1516).

Variables		Barriers to Oral Health Care			
		Yes	No	Total	*p*-Value
		N (%)	N (%)		
Region of residence	South	66 (25.1)	197 (74.9)	263 (17.3)	
	North	41 (31.3)	90 (68.7)	131 (8.6)	
	East	40 (22.0)	142 (78.0)	182(12.0)	0.272
	West	119 (25.7)	344 (74.3)	463 (30.5)	
	Central	108 (22.6)	369 (77.4)	477 (31.5)	
Child’s age	2–5	111 (19.3)	465 (80.7)	567 (38.0)	
	6–8	107 (26.1)	303 (73.9)	410 (27.0)	<0.001 *
	9–11	156 (29.4)	374 (70.6)	530 (35.0)	
Gender	Male	180 (23.0)	604 (77.0)	784 (51.7)	
	Female	194 (26.5)	538 (73.5)	732 (48.3)	0.110
Number of children ^t^ Mean ± SD		2.15 ± 1.352	2.15 ± 1.352	1.91 ± 1.91	<0.001
Maternal education	≤High school	151 (30.9)	338 (69.1)	489 (32.3)	<0.001 *
	≥University	223 (21.7)	804 (78.3)	1027 (67.7)	
Paternal education	≤High school	173 (33.5)	343 (66.5)	516 (34.0)	<0.001 *
	≥University	201 (20.1)	799 (79.9)	1000 (66.0)	
Total		374 (24.7)	1142 (75.3)	1516 (100)	

* Chi-square statistic was significant at the 0.05 level, ^t^
*t*-test for continuous variables.

**Table 2 ijerph-19-13584-t002:** Distribution of participants according to barriers to oral health care, parental description of their child’s oral health care needs, and oral manifestations (N = 1383).

Variables		Barriers to Oral Health Care			*p*-Value
		Yes	No	Total	
		N (%)	N (%)	N (%)	
How would you describe the health of your child’s teeth and gums in the last 12 months?	Excellent	28 (7.5)	356 (31.2)	384 (25.3)	<0.001
	Very Good	88 (23.5)	391 (34.2)	479 (31.6)	
	Good	162 (43.3)	320 (28.0)	482 (31.8)	
	Bad	83 (22.2)	57 (5.0)	140 (9.2)	
	I don’t know	13 (3.5)	18 (1.6)	31 (2.0)	
How often did your child have a toothache or feel discomfort due to his/her teeth in the last 12 months?	Always	45 (12.0)	31 (2.7)	76 (5.0)	<0.001
	Sometimes	194 (51.9)	293 (25.7)	487 (32.1)	
	Rarely	104 (27.8)	431 (37.7)	535 (35.3)	
	Never	24 (6.4)	353 (30.9)	377 (24.9)	
	I don’t know	7 (1.9)	34 (3.0)	41 (2.7)	
Have you detected any oral lesion/manifestations in your child’s mouth	Yes	250 (66.8)	347 (30.4)	597 (39.4)	<0.001
	No	124 (33.2)	795 (69.6)	919 (60.6)	
Oral lesions and manifestations detected by parents					
Pain while eating	Yes	80 (21.4)	95 (8.3)	175 (11.5)	<0.001 *
	No	294 (78.6)	1047 (91.7)	1341 (88.5)	
Teeth discoloration	Yes	87 (23.3)	107 (9.4)	194 (12.8)	<0.001 *
	No	287 (76.7)	1035 (90.6)	1322 (87.2)	
Bad odor/bad breath	Yes	105 (28.1)	126 (11.0)	231 (15.2)	<0.001 *
	No	269 (71.9)	1016 (89.0)	1285 (84.2)	
Swollen gums/swelling	Yes	33 (8.8)	39 (3.4)	72 (4.7)	<0.001 *
	No	341 (91.2)	1103 (96.6)	1444 (95.3)	
Dry mouth	Yes	10 (2.7)	12 (1.1)	22 (1.5)	0.023 *
	No	364 (97.3)	1130 (98.9)	1494 (98.5)	
Itchy gums	Yes	3 (0.8)	11 (1.0)	14 (0.9)	0.777 ^b^
	No	371 (99.2)	1131 (99.0)	1502 (99.1)	
Bleeding gums	Yes	43 (11.5)	38 (3.3)	81 (5.3)	<0.001 *
	No	331 (88.5)	1104 (96.7)	1435 (94.7)	
Difficulty in speaking	Yes	4 (1.1)	10 (0.9)	14 (0.9)	0.734 ^b^
	No	370 (98.9)	1132 (99.1)	1502 (99.1)	
Difficulty in eating	Yes	23 (6.1)	22 (1.9)	45 (3.0)	<0.001 *
	No	351 (93.9)	1120 (98.1)	1471 (97.0)	
Ulcer/multiple ulcers	Yes	14 (3.7)	14 (1.2)	28 (1.8)	0.002 *
	No	360 (96.3)	1128 (98.8)	1488 (98.2)	
White spots in the mouth or gums	Yes	31 (8.3)	54 (63.5)	85 (5.6)	0.009 *
	No	343 (24.0)	1088 (76.0)	1431 (94.4)	
Roughness in the skin or mouth	Yes	5 (41.7)	7 (0.6)	12 (0.8)	0.170 ^b^
	No	369 (91.7)	1135 (99.4)	1504 (99.2)	
Burning sensation	Yes	4 (1.1)	1 (0.1)	5 (0.3)	0.004 *^b^
	No	370 (98.9)	1141 (99.9)	1511 (99.7)	
Tumor	Yes	6 (1.60)	6 (0.5)	12 (0.8)	0.041 *^b^
	No	368 (98.4)	1136 (99.5)	1504 (99.2)	
Others	Yes	54 (14.4)	112 (9.8)	166 (10.9)	0.013 *
	No	320 (85.6)	1030 (90.2)	1350 (89.1)	

* Chi-square was significant at the 0.05 level; ^b^ Cell counts < 5; Fisher’s exact test may have been invalid.

**Table 3 ijerph-19-13584-t003:** Binary regression analysis showing the association between parents reporting barriers to dental care and sociodemographic factors.

Variables		AOR	95% CI	*p*-Value
Region of residence	South	1.079	0.749–1.556	0.682
	North	1.542	0.974–2.441	0.065
	East	0.937	0.605–1.452	0.771
	West	1.24	0.892–1.725	0.201
	Central region	1	-	-
Child’s age	3–5	0.631	0.462–0.862	0.004
	6–8	0.872	0.637–1.193	0.392
	9–11	1	-	-
Number of children		1.197	1.084–1.322	<0.001
Gender	Male	0.819	0.637–1.053	0.119
	Female	1	-	-
Maternal education	High school or less	1.151	0.853–1.552	0.359
	University or more	1	-	-
Paternal education	High school or less	1.815	1.368–2.409	<0.001
	University or more	1	-	-

Significance at *p* = 0.05.

**Table 4 ijerph-19-13584-t004:** Barriers to oral health care reported by parents (N = 1383).

Variables		N (%)
Fear of COVID-19	Yes	103 (6.6)
	No	1469 (93.4)
Distance from dental clinic	Yes	68 (4.3)
	No	1504 (95.7)
Appointment is expensive	Yes	156 (9.9)
	No	1416 (90.1)
Due to health ban	Yes	16 (1.0)
	No	1556 (99.0)
Difficulty of transportation	Yes	78 (5.0)
	No	1494 (95.0)
Responsibilities towards members of family	Yes	86 (5.5)
	No	1486 (94.5)
I don’t know how to make an appointment	Yes	12 (0.8)
	No	1560 (99.2)
No available appointment	Yes	154 (9.8)
	No	1418 (90.2)

## Data Availability

The data used in this study are available upon request from the corresponding author.

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
