# Peer review of "Oral Health Needs and Barriers among Children in Saudi Arabia"

_ijerph, 2022, doi:10.3390/ijerph192013584_

Round 1
Reviewer 1 Report
The study „Oral health needs and oral health care barriers among children in Saudi Arabia” deals with a relevant issue in paediatric dentistry. The study is well performed and reported, written in good English language. Regarding content and form, the paper is clear written and tables are lucid.
Nevertheless, there are a few concerns:
1.1. The selection of the included 1722 parents is not clear. Were parents of ALL children aged between 3 and 11 years living in those 4 geographical regions eligible to participate – and do they amount to 1772? Alternatively, was there some kind of selection modality?
22. Is there any reason, why authors split the children to regional groups? Are some regions more rural, others more urban, some better developed economically than others, are there different oral healthcare structures? If not, please consider to delete the numbers from the tables to increase the clearness of the structure. The finding, that there are no significant difference between the regions might be mentioned in the text of the results section.
33. Congratulations to that very high response rate. Usually, response rates of about 30% in online questionnaires are considered very good. What did the researchers do to reach such a high number of respondents? Which way of dissemination of the questionnaire was used?
44. Looking at the results (Tab.1), the proportion of parents from younger children accounting barriers was lower than the proportion of older children. So, older children met more barriers. This is an interesting finding and needs explanations in the discussion. Normally, the access for younger children is more difficult, as authors discuss. But their numbers report this issue vice versa.
55. The parents-reported toothache might have occurred due to physiological exfoliation of primary teeth (which might not need an appointment at the dentist in any case) in children over 6 years of age. Did authors do any efforts to reduce this observation bias?
66. It is very good to see in the result, that only 6,6% of the parents did not go to the dentist with their child due to fear of COVID19. This contrast to the work of Farsi shall be discussed further. What could be the reason for those differences? Perhaps, in the beginning of the pandemic, there was more uncertainty regarding the risk of infection, including within the dental setting.
77. Looking at the proportions again, it seems that the expense of the dental visit (9.9%) and the lacking availability of an appointment (9,8%) are the most frequently observed barriers. This interesting finding shall be discussed and explained in the discussion section. And then, the conclusion section needs an update – taking into account this results.
88. In table 4 the lines are presented twice. Please delete redundant lines.
99. Please state the SPSS Version.
Reviewer 2 Report
Thank you for submitting "Oral health needs and oral healthcare barriers among children in Saudi Arabia"
Perfect you have Ethics Comittee. Could you provide the original paper?
Perfect you validated the survey!!
Explains how the study sample size needed was arrived at.
How did you get in touch with those parents? Through social networks? through health centers? school campaigns?
The inclusion criteria included children >3 years old, however in the results there are children 2 years old.
In table 1, number of children, do you put yes or no as options, what does it refer to?
In my opinion, the results of the study are very debatable, because the parents do not give data about themselves (data that they are not able to give because they do not have dental training), in addition, they are about their children and they must have been reviewing their mouth constantly to find out if they have had any type of injuries specified in the survey.
Even though the manuscript was well written and the authors included well-described results and conclusions, the reviewer could not find the novelty of this work.
Reviewer 3 Report
Thank you for allowing me to review this paper. The purpose of the study was the present national study aimed to assess oral health needs and dental barriers among children aged 3–11 years after the start of the COVID-19 pandemic in Saudi Arabia.
The work is interesting, but certain refinements are needed:
1. Please provide the sample size calculation.
2. Some p values are rounded to three, and some to two decimal places. Please round everything to three decimal places.
3. In table 4, certain variables are repeated.
Round 2
Reviewer 1 Report
The concerns are explained and the manuscipt improved accordingly.
Author Response
-Thank you for your review and acceptance
Reviewer 2 Report
Even the manuscript was well written and the authors included well-described results and conclusions, the reviewer could not find the novelty of this work. It is up to the editor the decision.
Good luck.
Author Response
-Thank you for your review and comment. I hope this paper will have its impact through describing the oral health needs and barriers in Saudi Arabia.
Reviewer 3 Report
I thank the authors for noting and accepting the requested changes. The authors stated something about the calculation of the sample size (unclear to me), but they did not state how big that sample should be and whether it was satisfied by this study. In order for the manuscript to be accepted, the sample must be satisfactory.
Author Response
Thank you for your comment. The sample size was calculated as follow using OpenEpi:
|
Sample Size for Frequency in a Population |
||||||
|
|
||||||
|
Population size(for finite population correction factor or fpc)(N): |
1000000 |
|||||
|
Hypothesized % frequency of outcome factor in the population (p): |
30%+/-5 |
|||||
|
Confidence limits as % of 100(absolute +/- %)(d): |
5% |
|||||
|
Design effect (for cluster surveys-DEFF): |
1 |
|||||
|
Sample Size(n) for Various Confidence Levels |
||||||
|
|
||||||
|
Confidence |
Level(%) |
Sample Size |
||||
|
95% |
323 |
|||||
|
80% |
138 |
|||||
|
90% |
228 |
|||||
|
97% |
396 |
|||||
|
99% |
558 |
|||||
|
99.9% |
909 |
|||||
|
99.99% |
1271 |
|||||
|
|
||||||
|
Equation |
||||||
|
Sample size n = [DEFF*Np(1-p)]/ [(d2/Z21-α/2*(N-1)+p*(1-p)] |
||||||
Results from OpenEpi, Version 3, open source calculator--SSPropor
Print from the browser with ctrl-P
or select text to copy and paste to other programs.
The sample size was edited in the manuscript accordingly as follow:
“The suggested sample size for each region was 138 subjects; measured using OpenEpi (Version 3) online calculator, [22] with 95% confidence interval and a suggested prevalence of difficulty to access dental care of 29.2%; reported by children’s parents, according to Elkhodary et al 2022.
